

# A synthesis dataset of permafrost-affected soil thermal conditions for Alaska, USA

Kang Wang[1], Elchin Jafarov[2], Kevin Schaefer[3], Irina Overeem[1], Vladimir Romanovsky[4], Gary Clow[1,5], Frank Urban[5], William Cable[4,9], Mark Piper[1], Christopher Schwalm[6], Tingjun Zhang[7], Alexander Kholodov[4], Pamela Sousanes[8], Michael Loso[8], and Kenneth Hill[8]

[1]CSDMS, Institute of Arctic and Alpine Research, University of Colorado, Boulder, CO 80309, USA
[2]Los Alamos National Laboratory, Los Alamos, New Mexico, 87545, USA
[3]National Snow and Ice Data Center, Cooperative Institute for Research in Environmental Sciences, University of Colorado Boulder, Boulder, CO 80309, USA
[4]Geophysical Institute Permafrost Laboratory, University of Alaska, Fairbanks, AK 99775, USA
[5]U.S. Geological Survey, Lakewood, CO 80225, USA
[6]Woods Hole Research Center, Falmouth MA 02540, USA
[7]MOE Key Laboratory of Western China's Environmental Systems, College of Earth and Environmental Sciences, Lanzhou University, Lanzhou 730000, China
[8]National Park Service Arctic  Central Alaska Inventory and Monitoring Networks Fairbanks, AK 99709
[9]Alfred Wegener Institute Helmholtz Center for Polar and Marine Research, 14473 Potsdam, Germany

**Correspondence:** Kang Wang (Kang.Wang@colorado.edu)

**Abstract.** Recent observations of near-surface soil temperatures over the circumpolar Arctic show accelerated warming of permafrost-affected soils. A comprehensive near-surface permafrost temperature dataset is critical to better understand climate impacts and to constrain permafrost thermal conditions and spatial distribution in land system models. We compiled a soil temperatures dataset from 72 monitoring stations in Alaska using data collected by the U.S. Geological Survey, the National Park Service, and the University of Alaska-Fairbanks permafrost monitoring networks. The array of monitoring stations spans a large range of latitudes from 60.9°N to 71.3°N and elevations from near sea level to 1327 m, comprising tundra and boreal forest regions. This dataset consists of monthly ground temperatures at depth up to 1 m, volumetric soil water content, snow depth, and air temperature during 1997 - 2016. Due to the remoteness and harsh conditions, many stations have missing data. Overall, this dataset consists of 41,667 monthly values. These data have been quality controlled in collection and processing. Meanwhile, we implemented data harmonization validation for the processed dataset. The final product (PF-AK, v0.1) is available at the Arctic Data Center (https://doi.org/10.18739/A2KG55).

## 1 Introduction

Permafrost is frozen ground that remains at or below 0 °C for at least two consecutive years and may be found within about a quarter of the terrestrial land area in the Northern Hemisphere and 80% of the land area in Alaska (Brown et al., 1998; Zhang et al., 1999; Jorgenson et al., 2008). Continuous warming of the near-surface air temperatures over the Alaskan Arctic (Romanovsky et al., 2015; Wang et al., 2017) causes warming and thawing of permafrost in Alaska, which is expected to





continue throughout the 21st century with multi-billion dollar impacts on infrastructure and ecosystems (Callaghan et al., 2011; Hinzman et al., 2013; Liljedahl et al., 2016; Shiklomanov et al., 2017; Melvin et al., 2017). Permafrost thaw may have global consequences due to the potential for a significant positive climate feedback related to newly released carbon previously stored within the permafrost (Abbott et al., 2016; Schaefer et al., 2014; Knoblauch et al., 2018). Modeling studies

indicate that greenhouse gas emissions following thaw would amplify current rates of atmospheric warming (McGuire et al., 2016). However, large uncertainties exist regarding the timing and magnitude of this permafrost-carbon feedback, in part due to challenges associated with representation of permafrost processes in the climate models and the lack of comprehensive permafrost datasets with which to test such models (Koven et al., 2015; McGuire et al., 2016), There is an immediate need for ready-to-use reliable near-surface permafrost datasets, including ground temperatures, soil moisture, and related climatic

factors (such as air temperature and snow depth), which can serve as benchmarks for the modeling community and help to evaluate potential physical, societal, and economic impacts.

The permafrost extent map by Brown et al. (1998) is one of the most frequently and widely used metrics for comparing permafrost model results against real-world data (Koven et al., 2015; McGuire et al., 2016). Another widely used permafrost dataset is the Russian Soil Temperature dataset of daily ground temperature measurements at different depths ranging from 0

to 3.2 m for 51 years (Sherstiukov, 2012). An additional ground temperature dataset includes daily-mean ground temperatures at various depths from 0 to 3.2 m at more than 800 stations in China which in selected locations dates back to the 1950s (Wang et al., 2015).

A typical permafrost monitoring station consists of an air temperature sensor, a snow depth sensor, soil moisture sensors, and soil temperature sensors. In-situ observations of ground temperatures from the Alaskan Arctic region have been dispersed

over different monitoring efforts, which are spread over varying timespans, and have non-uniform depths. The maximum depth of a typical monitoring station ranges from 1 to 3 m below the ground surface. However, not all stations use this design. For example, the National Park Service of Alaska network does not collect soil moisture data. Also, data from permafrost monitoring stations in Alaska are not archived in a common standardized format and are hosted by different academic and government agencies, such as the Arctic Data Center, the Global Terrestrial Network for Permafrost (GTN-P), the Long Term

Ecological Research Network (LTER), and the U.S. Geological Survey (USGS). Thus, we compiled a ready-to-use permafrost dataset in order to of allow for efficient data retrieval and processing for permafrost-related analysis.

We compiled a first integrated near-subsurface ground temperatures dataset for permafrost-affected soils across Alaska from the three most reliable sources monitoring networks over several past decades: the Geophysical Institute Permafrost Laboratory at the University of Alaska Fairbanks (GI-UAF), National Park Services in Alaska (NPS), and the USGS. This

synthesis permafrost dataset for Alaska (PF-AK, version 0.1) includes measured air and ground temperatures, snow depth and soil volumetric water content for 72 permafrost monitoring stations across the state of Alaska. We provide detailed information and meta-data on the compiled dataset so that potential users can have a full understanding of the data and its associated limitations. Furthermore, we implemented two types of data harmonization validation: (i) we test for inconsistencies between air and ground temperature trends; and (ii) we use the snow heat transfer metric to validate the relations between seasonal





temperature amplitudes and snow depth. These technical validation would be useful for proving data harmonization and reusing these data.

## 2 Data sources and processing

### 2.1 Permafrost monitoring networks

Our synthesis permafrost dataset for Alaska is based on observed in-situ data collected by the USGS, NPS, and GI-UAF teams. In addition to these permafrost monitoring networks in Alaska, there is the Circumpolar Active Layer Monitoring (CALM) monitoring network measuring active layer thickness (ALT) - the maximum soil depth above permafrost that thaws every summer and refreezes in the winter. ALT is measured by physical probing on grids ranging in size of $100 \times 100$ to $1000 \times 1000$ meters, along specified transects, or from permanently installed frost tubes (Shiklomanov et al., 2008). Here we do not include

CALM data, since this particular data is already well organized, well documented, easily accessible, and does not require any significant data processing (Brown et al., 2000; Shiklomanov et al., 2008).

In the late 1990s, researchers at the GI-UAF established a near-surface permafrost monitoring system consisting of 27 stations across Alaska, primarily along the Trans-Alaskan highway (triangles in Fig.1)(Romanovsky et al., 2015). Similarly, the USGS installed permafrost stations to monitor permafrost conditions within the two federally managed areas on the North

Slope, the National Petroleum Reserve-Alaska and the Arctic National Wildlife Refuge. Since August 1998, the USGS has served 17 automated stations in the area spanning latitudes from 68.5°N to 70.5°N and longitudes from 142.5°W to 161°W (stars in Fig.1) (Urban and Clow, 2017). NPS has monitored permafrost conditions since 2004 (Hill and Sousanes, 2015). All monitoring stations are installed on undisturbed land (Fig.2) at a minimum specified distance from nearby infrastructure. This protocol for installation ensures no biases associated with anthropogenic or ecosystem disturbances, which is one of the main

differences with traditional meteorological stations which are often associated with airstrips and villages in Alaska. All the permafrost networks utilize radiation-shielded thermistors (Campbell Scientific CSI 107 temperature probes) to monitor air temperature. In the GI-UAF and NPS network, the air temperature sensors were installed at 1.5 or 2.0 m above the ground surface, whereas the USGS network monitors air temperature at 3.0 m above the ground surface in order to minimize damage by wildlife. Despite the installation of air temperature sensors at different heights, the measurements of air temperature are

considered comparable on a monthly scale assuming efficient mixing of the near-surface atmosphere.

To monitor near-surface ground temperature, the networks use either a probe with several thermistors embedded into a single rod, typically 1.0 to 1.5 m long, or several individual Campbell Scientific 107 thermistors anchored at specified depths within a single hole. The thermistor temperature sensors are designed to record temperatures ranging from -30 to 75 °C; the 107 sensors record temperatures from -35 to 50 °C. An ice-bath calibration is a required procedure before installation of these

30 probes. The ice-bath calibration includes placing the sensors into an insulated container filled with a mixture of ice shavings and distilled water, measuring the temperature, and recording the offset from 0 °C. This measured offset is then used to correct the temperature measurements. The average accuracy of these sensors is ±0.01 °C (Romanovsky et al., 2008). For the USGS network, the thermistor sensors are installed inside a tight-fitting fluid-filled 125-cm-long plastic tube to measure ground





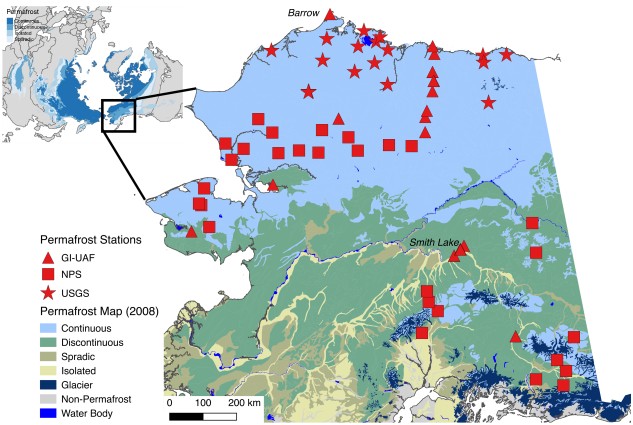

**Figure 1.** Locations of Geophysical Institute-University of Alaska Fairbanks (GI-UAF), U. S. Geological Survey (USGS), and National Park Services (NPS) permafrost monitoring stations in Alaska. The basemap is a new permafrost distribution of Alaska compiled by Jorgenson et al. (2008).

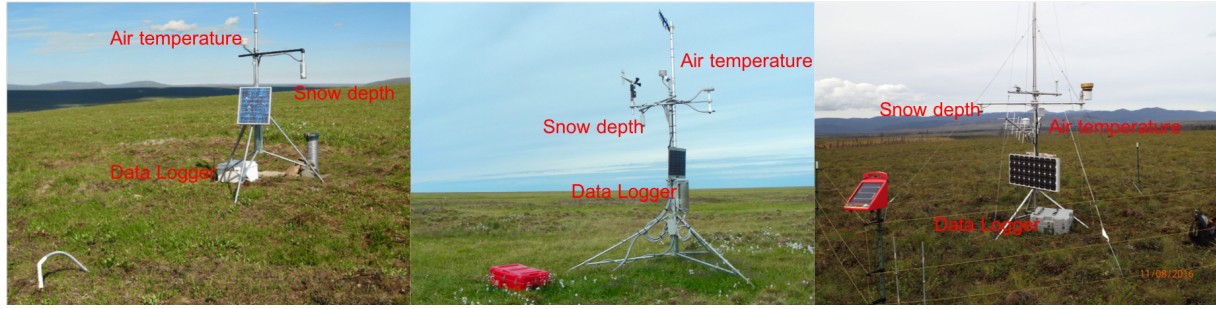

**Figure 2.** Typical permafrost observing stations. Left is Imnaviat 1 site (68.64 °N, 149.35°W) in the GI UAF network (source: http://permafrost.gi.alaska.edu/site/im1); middle is the Drew Point station (70.86°N, 153.91°W) in the USGS network (source:http://pubs.usgs.gov/ds/0977/DrewPoint/DrewPoint.html); right is the Wigand site (63.81°N, 150.109°W) in the NPS network.

temperatures at 5, 10, 15, 20, 25, 30, 45, 70, 95, and 120-cm depth (Urban and Clow, 2017). The NPS has three to four soil temperature sensors (CS1-107) installed in individual holes at 10, 20 and 50 cm depths, and at several locations an additional sensor at 100 cm. The ground-measurement depths vary station by station within the GI-UAF network, typically ranging from the ground surface (i.e., 0 m) to 1 m below the ground surface. It is important to note that most of the installed probes frost

5    heave with time, and heaving depths are adjusted accordingly by subtracting the heaving values yearly. The released data account for the heave and have corrected ground temperatures.

     Both the USGS and the GI-UAF networks measure liquid soil moisture using a Hydra Probe sensor developed by Stevens Water Monitoring Systems Inc. The Stevens Hydra Probe has a reported accuracy of $\pm 0.03 \ m^3/m^3$ (Bellingham, 2015). Each volumetric water content sensor was calibrated in accordance with the soil texture in laboratory while uncertainties associated

10   with the sensor's sensitivity still exist under specific conditions, e.g., for peat. The measured liquid soil moisture from a Hydro





Probe cannot be directly compared with the total soil moisture content values produced by land system models because in most of the models, soil moisture includes both ice and liquid water, where HydroProbe measures only liquid soil moisture. The USGS network measures soil moisture at one depth, approximately 0.15 m below the ground surface in all cases. The soil moisture sensors depths vary between stations for the GI-UAF network because they are installed at depths depending

on the soil profile and texture within the active layer. The GI-UAF network measures soil moisture typically at three different depths within the active layer, ranging from 0.10 to 0.60 m. The NPS network does not include moisture probes at any of their monitoring stations. Our processed dataset presents only the upper layer (up to 0.25 m) soil water content.

Snow depth is measured once per hour with a SR50 or SR50A ultrasonic distance sensor (Campbell Sci. Inc.) at all of the stations. This downward-looking sensor is mounted on a cross-arm typically at 2.5 m above the ground surface for the USGS

and NPS networks, and 1.5 m above the ground surface for the GI-UAF network respectively. The factory evaluated accuracy is ±0.01 m or 0.4% of the distance to the ground surface. It's important to note that vegetation at the ground surface might influence shallow snow-depth measurements.

## 2.2  Data processing workflow

All three networks apply data processing and quality-control (QC) checks before release. Typically, quality control occurs

shortly after annual summer field campaigns; the fully-processed and QC-ed data become publicly available a year after the data collection. In the present version of the permafrost dataset, we use USGS Data Series 1021, which includes data through July 2015 (URL: https://pubs.er.usgs.gov/publication/ds1021). The GI-UAF and NPS data were collected and processed by December, 20, 2017, and latest calibrated data was August, 2016. The GI-UAF data are available on http://permafrost.gi. alaska.edu/sites_map. NPS data is available from https://irma.nps.gov/DataStore/Reference/Profile/2240059 and https://irma.

nps.gov/DataStore/Reference/Profile/2239061.

Fig.3 shows a schematic representation of the data processing workflow used to compile the near-surface permafrost dataset. To standardize the ground temperature depth in the benchmarking dataset, we linearly interpolate ground temperatures for target depths: 0.25, 0.50, 0.75, and 1.00 m; however, we did not extrapolate beyond the deepest observed depth at any site. The USGS and NPS network releases data at hourly resolution, whereas the GI-UAF network releases data at daily resolution.

Since the most common model data output intervals of the land system and global climate models are monthly, we calculate monthly means of all measurements, including air and ground temperatures, snow depth, and soil water content. In addition to monthly data, we calculate annual means to allow evaluation of long-term trends between ambient and ground temperatures. Thus, the dataset also provides annual statistics including mean-annual air temperature (MAAT); mean-annual ground surface temperature (MAGST); mean-annual ground temperature at 1 m (MAGT at 1 m); mean and maximum seasonal snow depth

(SND); and maximum, mean, and minimum soil volumetric water content (VWC). We also calculated the Frost Number (i.e., Eq.1-3) for air temperature and ground temperatures following Nelson and Outcalt (1987). The Frost Number serves as a simplified index for the likelihood of permafrost occurrence.





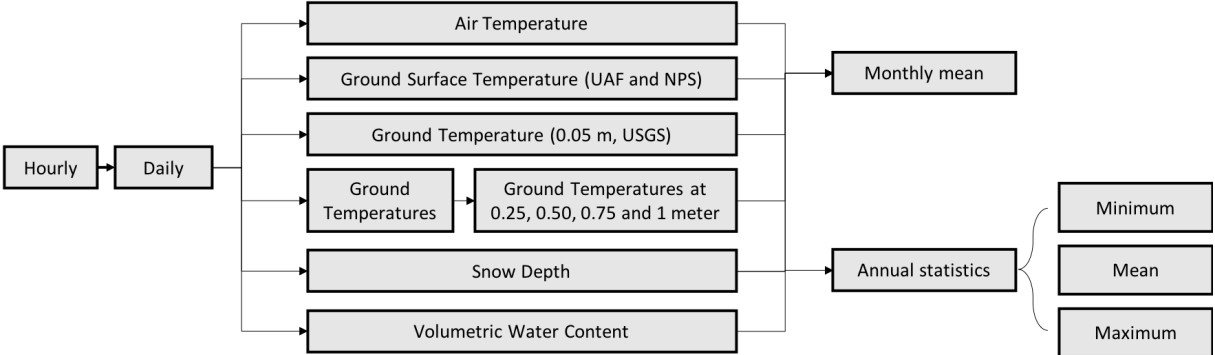

**Figure 3.** Schematic representation of the data processing workflow used to compile the permafrost dataset in the Alaska.

$$FrostNumber = \frac{\sqrt{DDF}}{\sqrt{DDF} + \sqrt{DDT}} \qquad (1)$$

$DDT$ and $DDF$ are given by

$$DDT = \int T(t)dt, T(t) > 0°C \qquad (2)$$

and

$$DDF = \int |T(t)|dt, T(t) \leq 0°C \qquad (3)$$

Data from many sites have gaps and discontinuities due to harsh environmental conditions and wildlife that may interrupt the monitoring. There are various methods for calculating monthly means from incomplete time series data. For example, the USGS standards allow only 5% of missing values for both monthly and annual mean temperature data (Urban and Clow, 2017). The World Meteorological Organization (WMO) does not allow gaps of more than three consecutive days or more than 5 days total from each monthly data series (Plummer et al., 2003). Other researchers are more tolerant of missing data, acknowledging the difficulty of data collection in remote cold regions. Menne et al. (2009) allows up to 10 missing days in a monthly time series. Bieniek et al. (2014) calculated monthly averages using at least 15 days. Here we calculated monthly means for any station which has at least 20 days of measurements for that specific month. The annual means were calculated from daily data. Due to the scarcity of the data, we calculate the annual means only for those years with a coverage of at least 90% of the daily data. For this reason, we present annual means for air and ground temperatures as well as soil moisture, derived from daily data.

During the dataset compilation, we identified similarly named sites with different installation times and locations that do not match precisely. It is important to note that these sites, even when located nearby each other, may have considerably different environmental conditions, and thus, different ground temperature thermodynamics. Our dataset allows only one name per



station site. We identified two overlapping sites in our new synthesis dataset: the Deadhorse site maintained by GI-UAF, and the Awuna site maintained by USGS. Both sites have new monitoring stations, and the old ones have been decommissioned. The environmental conditions for the newer Deadhorse station remained the same assuring data consistency. However, the environmental conditions between two monitoring stations at the Awuna site are quite different: the original Awuna site was

located on a ridge, whereas the new site is in a valley 1.9 km away. Nevertheless, the temperature data are consistent between the old and new station at the Awuna site. The old site (Awuna1) did not monitor soil moisture, which would be expected to be more site-specific and spatially variable. Thus, in this dataset, we present both the new and old sites' records.

## 2.3   Validation of data harmonization

Despite the fact that individual station observations had originally been quality-controlled, we still need to examine our own

results of the data harmonization. Here we implemented two ways of validation, the first way compares the trends in air and ground temperatures; and the second method examines the effects of snow on ground thermal states.

The primary objective of the trend analysis was to evaluate the consistency between trends at each station (for different depths) and between stations rather than inform inter-annual variability. Most of the estimated trends have a short observational period (see Tab.1). We chose to show trends only for those stations with more than five available annual means. Currently, some

of the time series are too short to provide significant trends. As more data becomes available in the future, a more rigorous analysis will be possible. It is well known that climatic trend analysis requires more than 30 years of time series (IPCC, 2013). On the other hand, Box et al. (2005) showed that 15 years are sufficient for inter-annual variability diagnosis to be statistically significant. Since the time series for most of the stations do not exceed 15 years we calculate trends for temperatures at different depths to determine inconsistencies between air and ground temperature trends in terms of sign's differences.

The second aspect is to examine the physical mechanism among air temperature, snow cover and ground thermal states, which is an auxiliary validation of the dataset. Seasonal snow cover will keep the ground warm by reducing the direct impact of cold air temperature during the winter (Yershov and Williams, 2004). Considering a semi-infinite column, the damping of the ground temperature annual cycle is depending on snow depth and thermal properties. In this study, the snow period was defined as October through March. To complement the snow thermal metric introduced by Slater et al. (2017) for Alaska, we

calculated the effective snow depth measurements ($SND_{eff}$) over the period from October through March. The temperature amplitudes of air ($Amp_{air}$) and ground surface ($Amp_{gnd}$) temperatures were calculated following Slater et al. (2017), which is calculated only for stations with available snow depth data. The snow and heat transfer metric (SHTM) was featured as the normalized temperature amplitude difference ($\Delta Amp_{norm}$) (i.e., Eq.4-6):

$$\Delta Amp_{norm} = \frac{Amp_{air} - Amp_{gnd}}{Amp_{air}} \tag{4}$$

$Amp_{air}$ and $Amp_{gnd}$ are given by

$$Amp_{air} = \frac{1}{2}[Max(T_{air}) - Min(T_{air})] \tag{5}$$



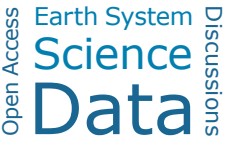

and

$$Amp_{gnd} = \frac{1}{2}[Max(T_{gnd}) - Min(T_{gnd})] \tag{6}$$

## 3 Results

### 3.1 Overview of this dataset

Tab.2 presents an overview of the data compiled in the permafrost benchmarking database for Alaska. Our dataset consists of a total of 41,667 monthly data values. The VWC shown in Tab.2 is from the upper part of the soil (i.e., up to 25 cm depth). The VWC measurements are mainly available on the North Slope of Alaska. Maximum VWC is more important for understanding active layer dynamics, especially during summer. Notably, the maximum VWC has a three times larger spatial variance than the annual means. Three sites, Chandalar Shelf, Pilgrim Hot Springs, and Red Sheep Creek, were much wetter than other sites

(maximum VWCs exceeding 0.7 $m^3/m^3$). This is mainly because these sites are close to a water body.

    Snow depth is spatially complex over Alaska, although with a general trend of increasing snow depth in the southern part of the state, according to the synthesis dataset (Fig.4). In the Alaskan Arctic, snow cover is shallower than in the southeast region. The highest maximum seasonal snow depth was 1.5 m at the Gates Glacier station (which is not actually located on the glacier) in Wrangell-St. Elias National Park. The lowest maximum snow depth occurs at WestDock near the Beaufort Sea in

Prudhoe Bay at only 0.09 m in 2010 (note that, there were available snow depth measurements only for 2010, i.e., we did not have enough data for any other years). Other two sites, Asik in Noatak National Park and Serpentine in Bering Land Bridge National Preserve, also showed a shallow snow cover in recent years (We don't intend to explain the reasons much more while they are mainly because of the topographical cause, e.g., Asik is a site on an exposed ridge).

    In this dataset, we derived the frost number index for air and ground temperatures at various depths (Fig.5 and Tab.3).

Because many stations do not have sensors at depth > 1 m, we report the freezing/thawing indices of air, ground surface, and 0.5 m below the ground surface in Fig.5, with all available results listed in Tab.3. Overall, almost all stations have air frost number above 0.5. Stations on the North Slope have both air and ground surface frost numbers exceeding 0.6. In interior and southern Alaska, air frost numbers were above 0.5, although the ground surface frost numbers were much lower due to the thicker snow cover in this region. In the Alaskan Arctic, thawing indices at ground surface were generally lower than air

according to the station observations. There were 13 stations with a zero thawing index of ground temperature at 0.5 m. These results indicate a shallow active layer (< 0.5 m) at these sites, which is consistent with the Active Layer Monitoring Network CALM data. Another five stations have a thawing index of ground temperature at 0.5 m less than 10 °C–days. The calculated frost number indices are consistent with the existing permafrost distribution map over the Alaska (Jorgenson et al., 2008).



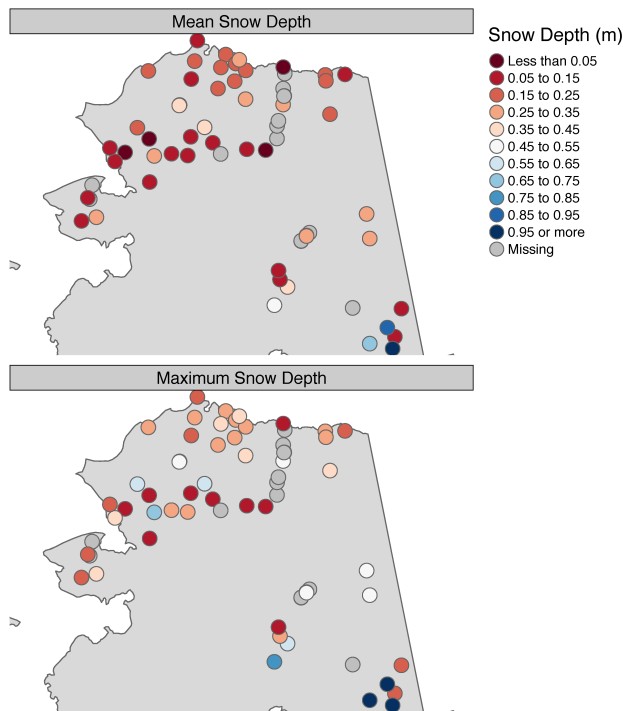

**Figure 4.** Overview of spatial distribution of snow depth, including annual mean snow depth and maximum snow depth.

### 3.2 Validation of data harmonization

We examined the consistency among the trends of MAAT, MAGST, and MAGT at 1 m depth. Typically, if MAAT has a long-term positive trend then MAGST is expected to have a positive trend, even if the rate is dampened (Romanovsky et al., 2015). Similarly, signs of trends in MAGST and MAGT at 1 m depth, and MAAT and MAGT at 1 m depth are hypothesized to be consistent (Romanovsky et al., 2015). Here we show the annual mean temperatures at four stations, Drew Point, Fish Creek, Niguanak, and Tunalik, with ten or more years of data (Fig.6). Mean annual air, ground surface, and ground temperature at 1 m indicates consistent warming at rates of 0.07 – 0.18, 0.14 – 0.23, and 0.12-0.22 °C/year, respectively. An obvious feature was that at Fish Creek, ground surface temperature and ground temperature at 1 m showed amplified warming rates compared to the magnitude of the air temperature increases, which can be explained by the significant increase of seasonal snow depth during the same period. Seven sites (Smith Lake 3, Ivotuk 4, GGLA2, Galbraith Lake, CTUA2, CREA2, and Chandalar Shelf) show inconsistent trends (Fig.7). However, when we considered the trend estimate uncertainties, we found that only two stations (Chandalar Shelf and Galbraith Lake) have significantly opposing trends between air temperature and ground surface temperature. Annual variation in snow depth may be responsible for the sign inconsistency between trends at these sites, but we lack observational evidence for this because snow depth data are not collected here.





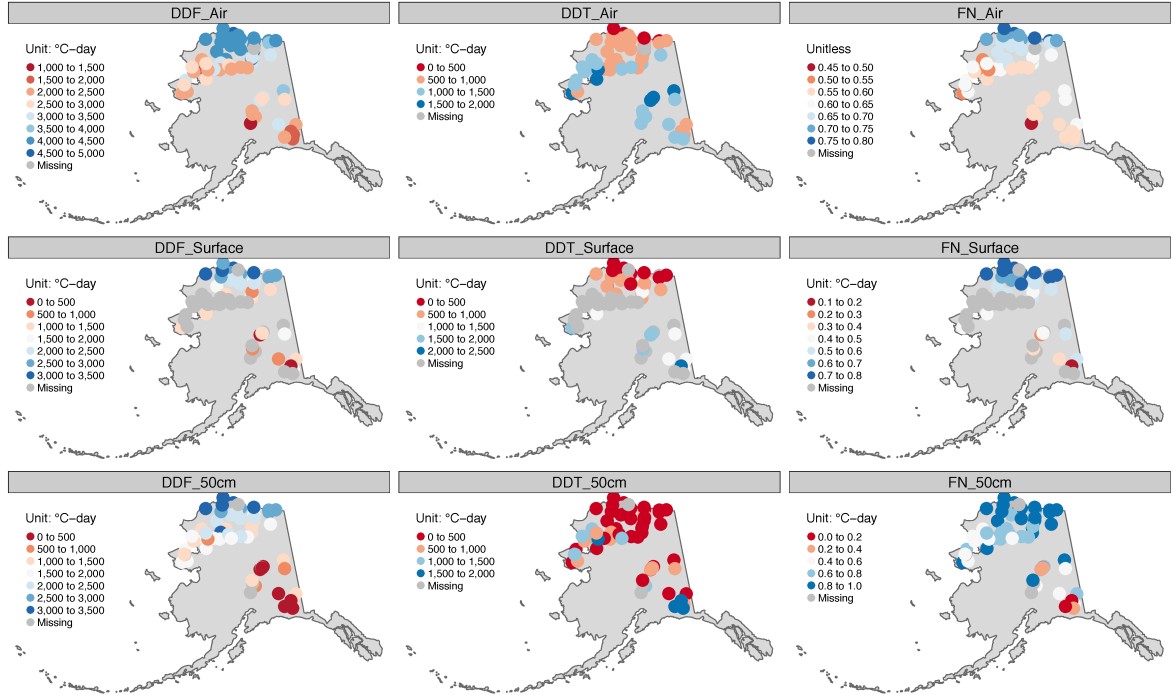

**Figure 5.** Overview of spatial distribution of freezing/thawing index from air, ground surface temperature, and ground temperature at 0.50 m. Frost Number (FN) was derived from the freezing/thawing index according to Nelson and Outcalt (1987).

Besides, there are several sites in a small area while indicated inconsistency in air temperature trends. This mainly because of different observational periods and relatively short duration of records. Typically, there are several Smith Lake (SL) permafrost monitoring stations which are located north of the UAF campus and west of Smith Lake with varying environmental conditions. (SL1 is in a White Spruce forest with high canopy; SL2 is in a dense diminutive Black Spruce forest; and SL3 is located at the edge of the forest surrounded by Black Spruce trees and tussock-shrubs; and SL4 is characterized by hummocks of sedges (tussocks) and shrubby vegetation with sparse Black Spruce.) The environmental conditions at SL3 site provide favorable conditions for permafrost existence. The SL3 site has the longest air temperature record indicating a cooling trend over the observational period (Fig.8A). After calculating the differences between measured data for all three sites we applied corresponding corrections and extend the data at all three sites. The overlap period (2006-2012) showed a consistent variation with the roughly constant offset between Smith Lake 2 and Smith Lake 3. By using the offset, we extended the records at Smith Lake 3 to 2015. Fig.8B shows that extending the time series reduces the trend magnitude and changes the negative sign in SL3 trend to positive, indicating the important difference between a complete versus a sparse time series.

Finally, we examined the physical relations among air temperature, snow cover and ground thermal states. We used the SHTM to examine the relationship between the effective snow depth and normalized temperature amplitude difference ($\Delta Amp_{norm}$) across the entire in-situ dataset. It should be noted that we lack sufficient observations (i.e., sites with snow, air temperature and



**Figure 6.** Examples of time-series in mean annual air, ground surface, ground temperature at 1 m below ground surface, and snow depth.





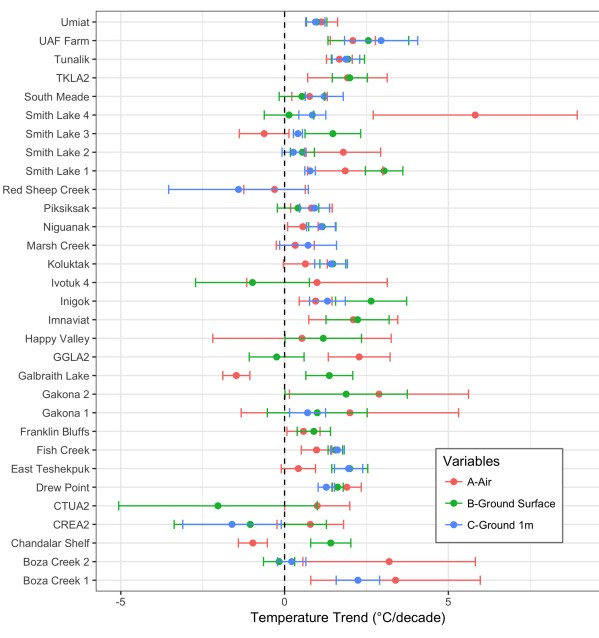

**Figure 7.** Trend comparison of air temperature, ground surface temperature, and ground temperature at 1 m. All trends were estimated only for the stations with ≥5-yr available data.

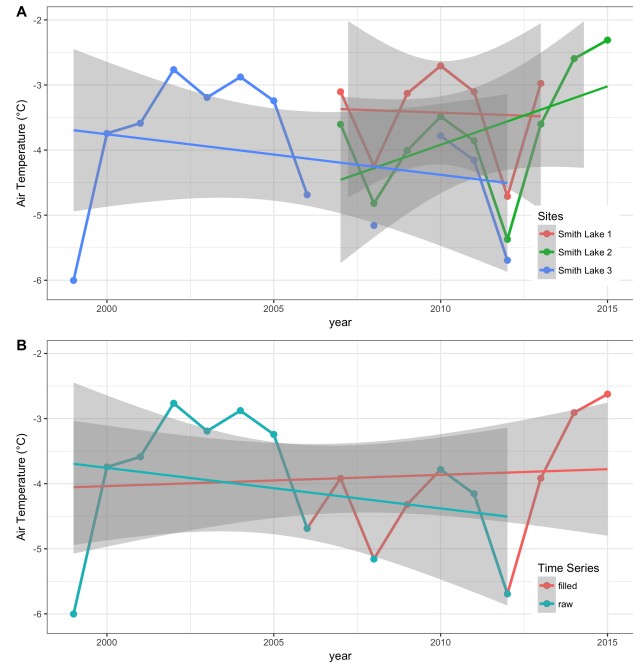

**Figure 8.** Comparison between A) trends calculated using measured data at Smith Lake 1,2,and 3; B) extended data and corrected trends at Smith Lake site 3.



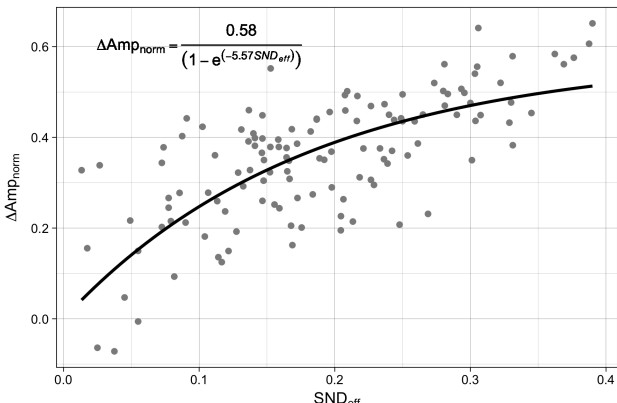

**Figure 9.** Correlation between effective snow depth and normalized temperature amplitude difference between air and ground surface. The mathematical function of fit line was following the correlation showed in Slater et al. (2017).

ground surface temperature) at sites with an effective snow depth over 0.5 m. Over the available observed range (snow depth < 0.4 m), SHTM suggests a positive and roughly linear relationship (Fig.9), implying snow insulation effects increase with increasing effective snow depth, which is consistent with previous studies(Burn and Smith, 1988; Demezhko and Shchapov, 2001; Zhang, 2005; Morse et al., 2012; Slater et al., 2017).

In this section, we presented two technical validations that are needed to support the technical quality and information justifying the reliability of these data. This information may help other researchers reuse this dataset.

## 4   Conclusions

Near-surface ground temperatures are important indicators of the rapidly warming Arctic, because they provide vital information on the response of the ground to climate change. In this paper, we describe the data compilation process listing the

work-flow and the challenges associated with preparing our synthesis permafrost dataset for Alaska. Standard unified protocols developed nationally and internationally to monitor near-surface permafrost conditions could significantly improve and simplify the development of corresponding permafrost benchmarks, and reduce the amount of time and effort required for data processing. This dataset consists of 41,667 monthly values during the data collection period (1997-2016). These data were quality-controlled in data collection and data processing stages. We also implemented data harmonization validation for this

compiled dataset. The PF-AK v0.1 can be easily integrated into model-data intercomparison tools such as International Land Model Benchmarking (ILAMB) tool (Luo et al., 2012). This dataset should be a valuable permafrost dataset and worth maintaining in the future. Widely, it also provides a prototype of basic data collection and management for remaining permafrost regions.



## 5   Data availability

The latest compiled dataset is available at the Arctic Data Center (https://doi.org/10.18739/A2KG55).

*Competing interests.*   The authors declare that they have no conflict of interest.

*Acknowledgements.*   This study was supported by the National Science Foundation (Award No. 1503559) and the NASA CMAC-14 project
5   (No. NNX16AB19G). GC and FU were supported by the U.S. Geological Survey's Climate and Land Use Change Program. National Park
Service data collection is supported by the NPS Inventory and Monitoring Program. GI-UAF Permafrost Lab data collection was supported
by the National Science Foundation (Awards OPP-0120736, ARC-0632400, ARC-0520578, ARC-0612533, and ARC-1304271) and by the
State of Alaska. TZ was supported by the National Natural Science Foundation of China (Award No. 91325202) and the National Key
Scientific Research Program of China (Award No. 2013CBA01802). We thank Dr. David Swanson for insightful comments and suggestions
10   on this manuscript. We also appreciate all for producing and making their data available. Any use of trade, firm, or product names is for
descriptive purposes only and does not imply endorsement by the U.S. Government.



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





**Table 1.** Overview of the data from the permafrost monitoring stations in Alaska

| Name | Latitude | Longitude | Onset | Last | MAAT | MAGST | MAGT 1 m | Snow Depth | Source |
|---|---|---|---|---|---|---|---|---|---|
| Awuna1 | 69.17 | -158.01 | 1998 | 2004 | 3 | 2 | 2 | 1 | USGS |
| Awuna2 | 69.16 | -158.03 | 2003 | 2015 | 7 | 1 | 1 | 5 | USGS |
| Camden Bay | 69.97 | -144.77 | 2003 | 2015 | 7 | | | | USGS |
| Drew Point | 70.86 | -153.91 | 1998 | 2015 | 11 | 12 | 12 | 8 | USGS |
| East Teshekpuk | 70.57 | -152.97 | 2004 | 2015 | 1 | 1 | 1 | 1 | USGS |
| Fish Creek | 70.34 | -152.05 | 1998 | 2015 | 14 | 15 | 15 | 11 | USGS |
| Itkpikpuk | 70.44 | -154.37 | 2005 | 2015 | 9 | 4 | | 5 | USGS |
| Inigok | 69.99 | -153.09 | 1998 | 2015 | 12 | 7 | 1 | 14 | USGS |
| Koluktak | 69.75 | -154.62 | 1999 | 2015 | 9 | 6 | 11 | | USGS |
| Lake 145Shore | 70.69 | -152.63 | 2007 | 2015 | 4 | | | 5 | USGS |
| Marsh Creek | 69.78 | -144.79 | 2001 | 2015 | 12 | 1 | 7 | 12 | USGS |
| Niguanak | 69.89 | -142.98 | 2000 | 2015 | 14 | 14 | 14 | 11 | USGS |
| Piksiksak | 70.04 | -157.08 | 2004 | 2015 | 1 | 7 | 1 | 8 | USGS |
| Red Sheep Creek | 68.68 | -144.84 | 2004 | 2015 | 7 | 1 | 6 | 7 | USGS |
| South Meade | 70.63 | -156.84 | 2003 | 2015 | 1 | 8 | 1 | 8 | USGS |
| Tunalik | 70.20 | -161.08 | 1998 | 2015 | 13 | 8 | 14 | 13 | USGS |
| Umiat | 69.40 | -152.14 | 1998 | 2015 | 14 | 13 | 13 | 11 | USGS |
| Barrow 2 | 71.31 | -156.66 | 2002 | 2016 | 4 | 9 | 6 | 4 | GI-U-AF |
| Boza Creek 1 | 64.71 | -148.29 | 2009 | 2016 | 6 | 1 | 6 | 5 | GI-U-AF |
| Boza Creek 2 | 64.72 | -148.29 | 2009 | 2016 | 6 | 6 | 6 | | GI-U-AF |
| Chandalar Shelf | 68.07 | -149.58 | 1997 | 2016 | 11 | 11 | | | GI-U-AF |
| Deadhorse | 70.16 | -148.47 | 1997 | 2016 | 3 | 3 | | | GI-U-AF |
| Fox | 64.95 | -147.62 | 2001 | 2016 | 3 | | | 4 | GI-U-AF |
| Franklin Bluffs | 69.67 | -148.72 | 1997 | 2016 | 13 | 1 | | | GI-U-AF |
| Franklin Bluffs boil | 69.67 | -148.72 | 2007 | 2016 | | 4 | | | GI-U-AF |
| Franklin Bluffs interior boil | 69.67 | -148.72 | 2006 | 2016 | | 6 | | 6 | GI-U-AF |
| Franklin Bluffs Wet | 69.68 | -148.72 | 2006 | 2016 | 3 | 3 | | | GI-U-AF |
| Galbraith Lake | 68.48 | -149.50 | 2001 | 2016 | 6 | 6 | | | GI-U-AF |
| Happy Valley | 69.16 | -148.84 | 2001 | 2016 | 6 | 8 | | 4 | GI-U-AF |
| Imnaviat | 68.64 | -149.35 | 2006 | 2016 | 8 | 8 | | | GI-U-AF |
| Ivotuk 3 | 68.48 | -155.74 | 2006 | 2013 | 2 | 2 | | | GI-U-AF |
| Ivotuk 4 | 68.48 | -155.74 | 1998 | 2016 | 6 | 5 | 1 | 6 | GI-U-AF |
| Pilgrim Hot Springs | 65.09 | -164.90 | 2012 | 2016 | 2 | 2 | 2 | 3 | GI-U-AF |
| Sag1 MNT | 69.43 | -148.67 | 2001 | 2016 | 7 | 3 | 1 | | GI-U-AF |
| Sag2 MAT | 69.43 | -148.70 | 2001 | 2016 | | 11 | 3 | 3 | GI-U-AF |
| Selawik Village | 66.61 | -160.02 | 2012 | 2016 | 3 | 3 | 3 | 3 | GI-U-AF |

| Name | Latitude | Longitude | Onset | Last | MAAT | MAGST | MAGT 1 m | Snow Depth | Source |
|---|---|---|---|---|---|---|---|---|---|
| Smith Lake 1 | 64.87 | -147.86 | 1997 | 2016 | 9 | 9 | 9 | | GI-UAF |
| Smith Lake 2 | 64.87 | -147.86 | 2006 | 2016 | 9 | 7 | 9 | | GI-UAF |
| Smith Lake 3 | 64.87 | -147.86 | 1997 | 2016 | 4 | 5 | 8 | | GI-UAF |
| Smith Lake 4 | 64.87 | -147.86 | 2006 | 2016 | 7 | 7 | 7 | | GI-UAF |
| UAF Farm | 64.85 | -147.86 | 2007 | 2016 | 7 | 6 | 5 | 4 | GI-UAF |
| West Dock | 70.37 | -148.55 | 2001 | 2016 | 9 | 4 | | 3 | GI-UAF |
| Gakona 1 | 62.39 | -145.15 | 2009 | 2016 | 5 | 5 | 5 | | GI-UAF |
| Gakona 2 | 62.39 | -145.15 | 2009 | 2016 | 5 | 5 | 3 | | GI-UAF |
| ASIA2 | 67.47 | -162.27 | 2012 | 2016 | 3 | | | 2 | NPS |
| CCLA2 | 65.31 | -143.13 | 2004 | 2016 | 11 | | | 8 | NPS |
| CHMA2 | 67.71 | -150.59 | 2012 | 2016 | 3 | | | 2 | NPS |
| CREA2 | 62.12 | -141.85 | 2004 | 2016 | 11 | 5 | 5 | 11 | NPS |
| CTUA2 | 61.27 | -142.62 | 2004 | 2016 | 11 | 5 | | 9 | NPS |
| DKLA2 | 63.27 | -149.54 | 2004 | 2016 | 9 | | 4 | 7 | NPS |
| DVLA2 | 66.28 | -164.53 | 2011 | 2016 | 4 | | | | NPS |
| ELLA2 | 65.28 | -163.82 | 2012 | 2016 | 3 | | | 1 | NPS |
| GGLA2 | 61.60 | -143.01 | 2005 | 2016 | 1 | 5 | | 5 | NPS |
| HOWA2 | 68.16 | -156.90 | 2011 | 2016 | 3 | | | 1 | NPS |
| IMYA2 | 67.54 | -157.08 | 2012 | 2016 | 3 | | | 1 | NPS |
| KAUA2 | 67.57 | -158.43 | 2012 | 2016 | 3 | | | 1 | NPS |
| KLIA2 | 67.98 | -155.01 | 2012 | 2016 | 2 | | | 1 | NPS |
| KUGA2 | 68.32 | -161.49 | 2014 | 2016 | 1 | | | | NPS |
| MITA2 | 65.82 | -164.54 | 2011 | 2016 | | | | | NPS |
| MNOA2 | 67.14 | -162.99 | 2011 | 2016 | 4 | | | 1 | NPS |
| PAMA2 | 67.77 | -152.16 | 2012 | 2016 | 2 | | | 2 | NPS |
| RAMA2 | 67.62 | -154.34 | 2012 | 2016 | 1 | | | | NPS |
| RUGA2 | 62.71 | -150.54 | 2008 | 2016 | 4 | | | 2 | NPS |
| SRTA2 | 65.85 | -164.71 | 2011 | 2016 | 4 | | | 3 | NPS |
| SRWA2 | 67.46 | -159.84 | 2001 | 2016 | 1 | | | 2 | NPS |
| SSIA2 | 68.00 | -160.40 | 2011 | 2016 | 4 | | | 2 | NPS |
| TAHA2 | 67.55 | -163.57 | 2011 | 2016 | 3 | | | 3 | NPS |
| TANA2 | 60.91 | -142.90 | 2005 | 2016 | 5 | | | 3 | NPS |
| TEBA2 | 61.18 | -144.34 | 2005 | 2016 | 8 | | | 6 | NPS |
| TKLA2 | 63.52 | -150.04 | 2005 | 2016 | 1 | 1 | | 8 | NPS |
| UPRA2 | 64.52 | -143.20 | 2005 | 2016 | 9 | 3 | | 4 | NPS |
| WIGA2 | 63.81 | -150.11 | 2013 | 2016 | 2 | | 5 | 1 | NPS |



**Table 2.** Summary of the air, ground surface, ground temperature at 1 m, volumetric water content and snow depth over the entire observation period.

| Site | Air Temperature (°C) Min | Mean | Max | Ground Surface Temperature (°C) Min | Mean | Max | Ground Temperature at 1 m (°C) Min | Mean | Max | VWC ($m^3/m^3$) Min | Mean | Max | Snow Depth (m) Mean | Max |
|---|---|---|---|---|---|---|---|---|---|---|---|---|---|---|
| Awuna1 | -28.51 | -10.61 | 9.62 | -11.30 | -4.16 | 2.79 | -9.38 | -4.52 | -0.93 | | | | 0.39 | 0.61 |
| Awuna2 | -30.47 | -9.88 | 11.60 | -13.21 | -3.34 | 8.10 | -10.84 | -4.43 | -0.64 | 0.02 | 0.21 | 0.43 | 0.37 | 0.54 |
| Camden Bay | -28.89 | -10.35 | 6.92 | | | | -14.47 | -7.49 | -1.20 | | | | 0.20 | 0.26 |
| Drew Point | -28.62 | -10.84 | 6.04 | -20.60 | -7.63 | 4.74 | -16.02 | -7.84 | -1.68 | | | | 0.18 | 0.29 |
| East Teshekpuk | -28.19 | -10.27 | 7.79 | -17.97 | -6.26 | 4.07 | -14.20 | -6.91 | -1.90 | 0.01 | 0.18 | 0.42 | 0.23 | 0.32 |
| Fish Creek | -29.07 | -10.55 | 8.81 | -16.85 | -6.02 | 4.50 | -14.11 | -6.82 | -1.17 | 0.01 | 0.17 | 0.41 | 0.20 | 0.28 |
| Ikpikpuk | -29.15 | -10.27 | 9.21 | -18.08 | -5.49 | 5.60 | | | | | | | 0.22 | 0.37 |
| Inigok | -29.98 | -10.58 | 10.55 | -16.28 | -4.80 | 7.73 | -12.68 | -5.58 | -0.60 | 0.00 | 0.12 | 0.33 | 0.22 | 0.33 |
| Koluktak | -30.02 | -10.18 | 11.64 | -15.20 | -3.77 | 8.75 | -13.77 | -4.69 | 1.16 | 0.02 | 0.13 | 0.36 | 0.20 | 0.30 |
| Lake145Shore | -28.72 | -10.50 | 7.30 | | | | | | | 0.06 | 0.21 | 0.41 | 0.28 | 0.42 |
| Marsh Creek | -26.51 | -8.65 | 10.20 | -16.87 | -5.28 | 5.26 | -14.39 | -6.11 | -0.82 | 0.03 | 0.16 | 0.41 | 0.19 | 0.25 |
| Niguanak | -27.80 | -9.97 | 8.48 | -18.13 | -6.09 | 4.66 | -14.87 | -6.72 | -1.02 | | | | 0.15 | 0.21 |
| Piksiksak | -29.21 | -9.93 | 10.71 | -17.65 | -5.76 | 6.21 | -13.44 | -5.94 | -0.87 | | | | 0.10 | 0.16 |
| Red Sheep Creek | -23.94 | -6.81 | 12.88 | -10.04 | -2.76 | 8.84 | -8.78 | -3.56 | -0.36 | 0.02 | 0.25 | 0.74 | 0.23 | 0.38 |
| South Meade | -29.90 | -10.42 | 9.35 | -19.91 | -6.45 | 5.89 | -15.74 | -7.19 | -1.12 | | | | 0.19 | 0.29 |
| Tunalik | -28.26 | -10.17 | 9.15 | -21.58 | -7.12 | 6.81 | -16.18 | -7.35 | -0.92 | | | | 0.17 | 0.28 |
| Umiat | -28.67 | -9.84 | 11.18 | -14.24 | -4.66 | 4.71 | -10.96 | -5.14 | -1.04 | | | | 0.32 | 0.44 |
| Barrow 2 | -26.55 | -10.23 | 5.09 | -19.17 | -6.87 | 5.33 | -15.46 | -7.41 | -1.59 | 0.02 | 0.16 | 0.39 | 0.14 | 0.22 |
| Boza Creek 1 | -25.00 | -3.20 | 16.03 | -9.17 | 1.13 | 12.93 | -4.58 | -1.27 | -0.29 | 0.00 | 0.20 | 0.55 | 0.18 | 0.36 |
| Boza Creek 2 | -23.60 | -2.18 | 16.31 | -3.62 | 2.28 | 12.00 | -0.46 | 0.09 | 1.23 | 0.06 | 0.22 | 0.40 | | |
| Chandalar Shelf | -23.66 | -7.64 | 11.41 | -9.54 | -1.29 | 7.74 | | | | 0.00 | 0.22 | 0.74 | | |
| Deadhorse | -28.04 | -9.97 | 8.27 | -14.89 | -3.65 | 7.13 | | | | 0.03 | 0.16 | 0.38 | | |
| Fox | -26.02 | -2.99 | 16.03 | | | | | | | 0.08 | 0.24 | 0.40 | | |
| Franklin Bluffs | -30.15 | -10.62 | 10.74 | -14.65 | -3.89 | 8.38 | | | | 0.02 | 0.19 | 0.47 | | |
| Franklin Bluffs boil | | | | -18.04 | -4.15 | 11.99 | | | | | | | | |
| Franklin Bluffs interior boil | | | | -16.85 | -3.66 | 11.12 | | | | | | | | |
| Franklin Bluffs Wet | -28.56 | -10.49 | 10.84 | -14.52 | -3.36 | 10.28 | | | | | | | | |
| Galbraith Lake | -28.77 | -9.35 | 10.72 | -14.38 | -3.45 | 9.34 | | | | | | | | |
| Happy Valley | -30.01 | -9.49 | 12.30 | -9.31 | -1.63 | 7.19 | | | | 0.02 | 0.14 | 0.31 | 0.27 | 0.47 |
| Imnaviat | -22.95 | -6.81 | 10.57 | -8.48 | -0.81 | 8.54 | | | | | | | | |
| Ivotuk 3 | -29.85 | -10.12 | 11.30 | -9.97 | -1.14 | 6.99 | | | | | | | | |
| Ivotuk 4 | -29.10 | -9.70 | 11.23 | -9.21 | -1.24 | 8.26 | -5.16 | -1.89 | -0.53 | 0.00 | 0.27 | 0.77 | 0.43 | 0.60 |
| Pilgrim Hot Springs | -16.78 | -2.04 | 14.63 | -11.95 | 0.08 | 13.52 | -7.56 | -2.30 | -0.27 | 0.00 | 0.30 | 0.73 | 0.06 | 0.21 |
| Sag1 MNT | -26.72 | -8.39 | 10.68 | -17.14 | -4.27 | 9.48 | -13.50 | -5.00 | 0.24 | 0.04 | 0.20 | 0.40 | | |
| Sag2 MAT | | | | -15.11 | -3.76 | 9.01 | -11.03 | -4.49 | -0.45 | 0.02 | 0.26 | 0.63 | | |
| Selawik Village | -20.26 | -3.72 | 14.91 | -11.16 | -0.74 | 12.18 | -7.99 | -3.09 | -0.45 | | | | 0.05 | 0.12 |





**Table 2.** Summary of the air, ground surface, ground temperature at 1 m, volumetric water content and snow depth over the entire observation period—continued.

| Site | Air Temperature (°C) Min | Mean | Max | Ground Surface Temperature (°C) Min | Mean | Max | Ground Temperature at 1 m (°C) Min | Mean | Max | VWC ($m^3/m^3$) Min | Mean | Max | Snow Depth (m) Mean | Max |
|---|---|---|---|---|---|---|---|---|---|---|---|---|---|---|
| Smith Lake 1 | -23.88 | -3.06 | 16.06 | -11.29 | -0.11 | 12.98 | -2.02 | -0.73 | -0.26 | 0.02 | 0.14 | 0.31 | | |
| Smith Lake 2 | -24.91 | -3.74 | 15.98 | -7.32 | 1.10 | 12.86 | -4.10 | -1.11 | 0.00 | 0.07 | 0.29 | 0.59 | | |
| Smith Lake 3 | -27.29 | -4.70 | 14.68 | -3.49 | 2.57 | 11.51 | -0.33 | 0.00 | 0.88 | 0.07 | 0.23 | 0.40 | | |
| Smith Lake 4 | -26.15 | -3.58 | 18.20 | -15.81 | -2.27 | 9.68 | -10.32 | -3.81 | -0.62 | | | | | |
| UAF Farm | -22.09 | -1.48 | 16.57 | -10.91 | 0.68 | 13.00 | -0.83 | 1.18 | 5.43 | | | | 0.28 | 0.47 |
| West Dock | -28.82 | -10.53 | 6.81 | -20.30 | -6.68 | 5.46 | | | | 0.01 | 0.20 | 0.55 | 0.04 | 0.09 |
| Gakona 1 | -23.06 | -2.76 | 13.70 | -5.29 | 1.55 | 11.26 | -1.62 | -0.63 | -0.22 | | | | | |
| Gakona 2 | -23.01 | -2.45 | 14.00 | -5.54 | 1.35 | 9.63 | -0.72 | -0.18 | 0.75 | | | | | |
| ASIA2 | -15.10 | -3.20 | 12.24 | | | | | | | | | | 0.02 | 0.07 |
| CCLA2 | -27.39 | -4.52 | 15.90 | | | | | | | | | | 0.33 | 0.52 |
| CHMA2 | -15.97 | -5.24 | 9.81 | | | | | | | | | | 0.04 | 0.08 |
| CREA2 | -16.41 | -3.87 | 8.57 | -12.35 | -1.78 | 11.22 | -6.00 | -2.13 | 0.35 | | | | 0.12 | 0.21 |
| CTUA2 | -14.15 | -2.52 | 8.61 | -12.83 | -1.09 | 12.43 | | | | | | | 0.08 | 0.16 |
| DKLA2 | -17.19 | -3.32 | 10.72 | | | | -3.33 | 1.22 | 7.03 | | | | 0.39 | 0.64 |
| DVLA2 | -21.84 | -5.38 | 10.77 | | | | | | | | | | | |
| ELLA2 | -17.18 | -4.81 | 9.93 | | | | | | | | | | 0.29 | 0.43 |
| GGLA2 | -13.51 | -2.01 | 9.13 | -1.50 | 2.54 | 12.18 | | | | | | | 0.90 | 1.45 |
| HOWA2 | -23.29 | -6.64 | 10.18 | | | | | | | | | | 0.05 | 0.11 |
| IMYA2 | -15.30 | -5.19 | 8.96 | | | | | | | | | | 0.15 | 0.26 |
| KAUA2 | -21.65 | -6.47 | 10.01 | | | | | | | | | | 0.15 | 0.25 |
| KLIA2 | -19.10 | -7.66 | 7.38 | | | | | | | | | | 0.07 | 0.10 |
| KUGA2 | -16.74 | -3.56 | 13.64 | | | | | | | | | | 0.18 | 0.59 |
| MITA2 | | | | | | | | | | | | | | |
| MNOA2 | -18.78 | -3.79 | 12.47 | | | | | | | | | | 0.14 | 0.37 |
| PAMA2 | -18.00 | -4.49 | 11.02 | | | | | | | | | | 0.07 | 0.11 |
| RAMA2 | -17.93 | -5.42 | 10.77 | | | | | | | | | | | |
| RUGA2 | -9.49 | -0.53 | 10.45 | | | | | | | | | | 0.50 | 0.83 |
| SRTA2 | -21.96 | -4.69 | 11.77 | | | | | | | | | | 0.06 | 0.16 |
| SRWA2 | -17.35 | -3.15 | 13.89 | | | | | | | | | | 0.34 | 0.68 |
| SSIA2 | -21.85 | -5.86 | 11.27 | | | | | | | | | | 0.02 | 0.06 |
| TAHA2 | -20.09 | -4.48 | 11.58 | | | | | | | | | | 0.09 | 0.20 |
| TANA2 | -13.83 | -2.02 | 9.91 | | | | | | | | | | 1.01 | 1.55 |
| TEBA2 | -17.27 | -1.92 | 11.54 | | | | | | | | | | 0.75 | 1.34 |
| TKLA2 | -18.48 | -3.15 | 11.39 | -6.93 | 1.63 | 13.17 | | | | | | | 0.15 | 0.25 |
| UPRA2 | -21.39 | -4.91 | 11.36 | -13.19 | -1.69 | 12.80 | | | | | | | 0.33 | 0.48 |
| WIGA2 | -17.84 | -1.55 | 13.21 | | | | | | | | | | 0.10 | 0.15 |





**Table 3.** Summary of the freezing (DDF) and thawing index (DDT) of air and ground temperatures over the entire observation period (unit: $°C − day$).

| Site | Air DDF | Air DDT | Ground Surface DDF | Ground Surface DDT | Ground 0.25 m DDF | Ground 0.25 m DDT | Ground 0.50 m DDF | Ground 0.50 m DDT | Ground 0.75 m DDF | Ground 0.75 m DDT | Ground 1.00 m DDF | Ground 1.00 m DDT |
|---|---|---|---|---|---|---|---|---|---|---|---|---|
| Awuna1 | 4217 | 769 | 1750 | 196 | 1862 | 10 | 1878 | 0 | 1880 | 0 | 1880 | 0 |
| Awuna2 | 4417 | 975 | 1740 | 807 | 1939 | 233 | 2086 | 7 | 2121 | 0 | 2095 | 0 |
| Camden Bay | 4493 | 482 | | | 2684 | 100 | 2858 | 0 | 2873 | 0 | 2860 | 0 |
| Drew Point | 4521 | 400 | 3221 | 327 | 3291 | 46 | 3280 | 0 | 3248 | 0 | 3231 | 0 |
| East Teshekpuk | 4298 | 576 | 2815 | 279 | 2964 | 18 | 2982 | 0 | 2951 | 0 | 2939 | 0 |
| Fish Creek | 4376 | 677 | 2582 | 328 | 2813 | 12 | 2821 | 0 | 2804 | 0 | 2789 | 0 |
| Ikpikpuk | 4356 | 718 | 2712 | 434 | 2685 | 225 | | | | | | |
| Inigok | 4404 | 858 | 2268 | 708 | 2454 | 60 | 2491 | 0 | 2449 | 0 | 2423 | 0 |
| Koluktak | 4337 | 984 | 2034 | 856 | 2242 | 618 | 2309 | 325 | 2340 | 153 | 2355 | 54 |
| Lake145Shore | 4430 | 522 | | | | | | | | | | |
| Marsh Creek | 3836 | 860 | 2526 | 408 | 2831 | 159 | 2863 | 20 | 2801 | 0 | 2776 | 0 |
| Niguanak | 4179 | 654 | 2798 | 339 | 2952 | 54 | 2960 | 1 | 2934 | 0 | 2900 | 0 |
| Piksiksak | 4263 | 886 | 2594 | 506 | 2700 | 66 | 2707 | 0 | 2657 | 0 | 2611 | 0 |
| Red Sheep Creek | 3249 | 1230 | 1208 | 989 | 1637 | 324 | 1715 | 58 | 1710 | 0 | 1667 | 0 |
| South Meade | 4477 | 727 | 3006 | 447 | 3186 | 45 | 3214 | 0 | 3187 | 0 | 3078 | 0 |
| Tunalik | 4213 | 725 | 3230 | 535 | 3258 | 138 | 3225 | 8 | 3160 | 0 | 3120 | 0 |
| Umiat | 4138 | 948 | 2114 | 374 | 2306 | 14 | 2271 | 0 | 2216 | 0 | 2189 | 0 |
| Barrow 2 | 4241 | 325 | 2925 | 398 | 2996 | 85 | 3072 | 0 | 3049 | 0 | 3112 | 0 |
| Boza Creek 1 | 3270 | 1634 | 959 | 1646 | 676 | 581 | 832 | 81 | 917 | 1 | 888 | 0 |
| Boza Creek 2 | 3036 | 1704 | 278 | 1808 | 224 | 839 | 166 | 550 | 103 | 308 | 47 | 188 |
| Chandalar Shelf | 3285 | 1049 | 1184 | 855 | 1352 | 55 | 1302 | 0 | 1388 | 0 | | |
| Deadhorse | 4236 | 628 | 2070 | 654 | 2106 | 261 | 2144 | 101 | 2236 | 3 | | |
| Fox | 3441 | 1618 | | | 192 | 442 | 214 | 21 | 191 | 0 | | |
| Franklin Bluffs | 4420 | 879 | 1964 | 820 | 2096 | 237 | 2114 | 61 | 2289 | 1 | | |
| Franklin Bluffs boil | | | 2339 | 1234 | 2293 | 792 | 2117 | 414 | 2018 | 193 | | |
| Franklin Bluffs interior boil | | | 2192 | 1145 | 2132 | 498 | 2166 | 288 | 2073 | 111 | | |
| Franklin Bluffs Wet | 4142 | 907 | 1873 | 1100 | 1733 | 635 | 1734 | 689 | 1702 | 68 | | |
| Galbraith Lake | 4190 | 895 | 1875 | 955 | 2050 | 167 | 2110 | 14 | 2123 | 0 | | |
| Happy Valley | 4293 | 1061 | 1167 | 781 | 1245 | 211 | 1337 | 36 | 1404 | 0 | | |
| Imnaviat | 3212 | 954 | 994 | 1005 | 1017 | 460 | 1053 | 218 | 1086 | 93 | | |
| Ivotuk 3 | 4332 | 948 | 1273 | 729 | 1134 | 127 | 1312 | 3 | 1312 | 0 | | |
| Ivotuk 4 | 4209 | 948 | 1105 | 933 | 1142 | 579 | 1248 | 120 | 1290 | 6 | 1038 | 0 |
| Pilgrim Hot Springs | 2025 | 1632 | 1346 | 1631 | 1723 | 168 | 1583 | 18 | 1465 | 1 | 1427 | 0 |
| Sag1 MNT | 3840 | 912 | 2313 | 914 | 2209 | 521 | 2227 | 202 | 2259 | 36 | 2425 | 5 |
| Sag2 MAT | | | 2012 | 900 | 2207 | 186 | 2287 | 44 | 2281 | 12 | 2098 | 3 |
| Selawik Village | 2556 | 1579 | 1266 | 1452 | 1626 | 148 | 1695 | 0 | 1608 | 0 | 1542 | 0 |



**Table 3.** Summary of the freezing and thawing index of air and ground temperatures over the entire observation period (unit: $^{\circ}C - day$)—continued.

| Site | Air DDF | Air DDT | Ground Surface DDF | Ground Surface DDT | Ground 0.25 m DDF | Ground 0.25 m DDT | Ground 0.50 m DDF | Ground 0.50 m DDT | Ground 0.75 m DDF | Ground 0.75 m DDT | Ground 1.00 m DDF | Ground 1.00 m DDT |
|---|---|---|---|---|---|---|---|---|---|---|---|---|
| Smith Lake 1 | 3086 | 1659 | 1273 | 1581 | 488 | 70 | 469 | 1 | 429 | 0 | 415 | 0 |
| Smith Lake 2 | 3254 | 1624 | 712 | 1723 | 779 | 392 | 810 | 120 | 781 | 13 | 748 | 1 |
| Smith Lake 3 | 3703 | 1403 | 275 | 1739 | 227 | 773 | 114 | 514 | 60 | 324 | 36 | 137 |
| Smith Lake 4 | 3384 | 1934 | 2084 | 966 | 1815 | 353 | 2064 | 39 | 2082 | 0 | 1996 | 0 |
| UAF Farm | 2779 | 1773 | 1216 | 1599 | 499 | 1043 | 279 | 959 | 135 | 949 | 51 | 891 |
| West Dock | 4491 | 475 | 3108 | 400 | 3181 | 22 | 3186 | 0 | 3121 | 0 | | |
| Gakona 1 | 3068 | 1361 | 483 | 1573 | 434 | 303 | 443 | 35 | 437 | 0 | 336 | 0 |
| Gakona 2 | 3046 | 1402 | 564 | 1311 | 428 | 578 | 261 | 294 | 160 | 233 | 139 | 145 |
| ASIA2 | 1861 | 1339 | | | 1657 | 1150 | 1617 | 1030 | | | | |
| CCLA2 | 3656 | 1559 | | | 1430 | 551 | 1162 | 23 | 1113 | 3 | | |
| CHMA2 | 2104 | 981 | | | 2222 | 936 | 1837 | 478 | 1537 | 358 | | |
| CREA2 | 2248 | 817 | 1481 | 1274 | 1412 | 725 | 1267 | 396 | 1131 | 129 | 1046 | 15 |
| CTUA2 | 1880 | 868 | 1510 | 1438 | 1434 | 870 | 1310 | 751 | | | | |
| DKLA2 | 2264 | 1084 | | | 725 | 1350 | 566 | 1216 | 428 | 1098 | 321 | 997 |
| DVLA2 | 3031 | 1010 | | | 1742 | 360 | 1724 | 143 | | | | |
| ELLA2 | 2298 | 975 | | | 1545 | 1030 | 1530 | 760 | | | | |
| GGLA2 | 1753 | 953 | 79 | 2028 | 17 | 1824 | 4 | 1642 | | | | |
| HOWA2 | 3292 | 901 | | | 3295 | 678 | 3111 | 516 | | | | |
| IMYA2 | 2038 | 880 | | | 1849 | 995 | 1887 | 547 | | | | |
| KAUA2 | 3027 | 904 | | | 1764 | 623 | 1674 | 452 | | | | |
| KLIA2 | 2763 | 624 | | | 2201 | 366 | 2257 | 208 | | | | |
| KUGA2 | 2057 | 1491 | | | 1255 | 1418 | 1245 | 1066 | | | | |
| MITA2 | | | | | | | | | | | | |
| MNOA2 | 2447 | 1295 | | | 963 | 1050 | 1144 | 959 | 1059 | 704 | | |
| PAMA2 | 2374 | 1101 | | | 2135 | 611 | 2117 | 409 | | | | |
| RAMA2 | 2373 | 1066 | | | 1916 | 952 | 1854 | 1036 | | | | |
| RUGA2 | 1075 | 1250 | | | | | | | | | | |
| SRTA2 | 2998 | 1138 | | | 1192 | 1147 | 1063 | 1122 | | | | |
| SRWA2 | 2142 | 1510 | | | 928 | 1826 | 786 | 1516 | | | | |
| SSIA2 | 2993 | 1062 | | | 2234 | 771 | 2165 | 608 | 1789 | 526 | | |
| TAHA2 | 2702 | 1149 | | | 1590 | 1175 | 1565 | 1027 | 1399 | 631 | | |
| TANA2 | 1770 | 1053 | | | 171 | 1850 | 106 | 1505 | | | | |
| TEBA2 | 2237 | 1191 | | | 66 | 1985 | 28 | 1757 | | | | |
| TKLA2 | 2446 | 1151 | 669 | 1809 | | | | | | | | |
| UPRA2 | 2913 | 1083 | 1552 | 1481 | 1084 | 1142 | 884 | 832 | | | | |
| WIGA2 | 2246 | 1402 | | | 1053 | 289 | 1120 | 59 | | | | |