# Peer review of "A synthesis dataset of permafrost-affected soil thermal conditions for Alaska, USA"

_Earth System Science Data, 2018_

## Referee Comment (RC1) · Anonymous Referee #1 · 7 Jun 2018

Comments to the Authors

The manuscript describes a dataset of thermal conditions for permafrost-affected soils in Alaska. This complements an earlier published dataset providing deep ground temperature that was described by Clow (2013, ESSD). The compilation could be described as a value-added dataset which might be a more preferable term than synthesis dataset. The authors have gone beyond providing simply a compilation of raw data acquired from many sites and have calculated a number of key parameters and statistics. This value-added compilation will be useful to permafrost scientists, ecologists, hydrologists, engineers and practitioners as well as the modelling community. The manuscript provides a detailed description of the development of the dataset including the various processing steps and techniques used for quality control. It provides useful

information that might serve as guidance for others that are compiling similar types of data for public dissemination. For these reasons, this manuscript should be published.

The manuscript however, requires a bit of work before it is acceptable for publication. There are a few places in the manuscript where more explanation would be helpful. For example, some of the parameters utilized such as effective snow depth and SHTM require further explanation (see specific comments). Other sections that require further explanation are outlined in the specific comments. For the most part the manuscript is well written but some editing is required to improve language and increase clarity. Some suggestions for editorial revisions have been provided but the authors should thoroughly proofread the revised manuscript before submission.

Although I have made several comments on the manuscript that I hope the authors will find helpful, dealing with them should not take much time. I expect that a revised manuscript that is acceptable for publication can be prepared within a reasonable time. I look forward to reading the published paper.

Specific Comments (keyed to page and line numbers)

P1,L1 – This is not a conclusion of this paper so it could be deleted. P1,L15 – It is better to use "increasing" rather than "warming" when referring to temperature. Suggested revision "Continuous increases in near-surface air temperatures. . ." or alternatively you could say "Continuous warming at the ground surface. . .." P2,L1 – Are you placing a dollar value on ecosystems? P2,L13-17 – Are these really permafrost datasets or is soil temperature (or shallow ground temperature) dataset a better description given that the measurements may not necessarily be in permafrost.

P2,L18-26 - There are other permafrost monitoring sites in Alaska and perhaps these should be mentioned. There are the measurements to about 20m that UAF collects and also the deeper temperatures collected by the USGS which have been published (see Clow 2013, ESSD). These could also be mentioned either here or in previous paragraph. P2, L27 – "near-surface ground temperatures" or "shallow ground temperatures" might be better terminology.

P2,L28 – revision suggested "…..from the three most reliable monitoring networks over the past several decades:…." P2,L30 – indicate at what depth the ground temperatures are measured, i.e. "…ground temperatures to x depth)…" P2,L31 – revision suggested "…..for 72 stations…." P2,L31-34 – Consider reducing the use of first person. Eg. "Detailed information and meta-data are provided for the dataset…" "Futhermore, two types of data …..…were implemented: (i) testing for inconsistencies…....; and (ii)…..use of the snow….."

P3,L6-9 – I don't think you need to give the description of the CALM network as these data are not compiled in the dataset that is the subject of this paper. I suggest that this section be deleted. You can mention in the Introduction that the dataset you have compiled complements other permafrost relevant datasets compiled for AK such as CALM and USGS (see above) datasets. The focus in this section should only be a description of the sources for your data compilation.

P3, L13 – In figure reference (here and elsewhere) you can remove the symbol as this information should be in the figure caption or legend.

P3, L14 – revision "…..USGS installed stations to monitor permafrost…" P3, L16 – revision "…..the USGS has maintained 17 automated…" P3, L17 – is "NPS has monitored ground temperatures since 2004" more appropriate?

P3, L26 – P4,L6 – There is some repetition in this section and it is a bit confusing. You could say that thermistors are utilized to measure temperatures to depths of 1.5m and that these are embedded in a rod, anchored in a single hole or inside a fluid-filled hole. You could then describe the calibration procedure and give the accuracy (should also give precision). The details of the systems used in the 3 networks including thermistor type and temperature range, measurement depths and installation method could then be summarized in a table along with any relevant publications for the particular network. The data acquisition system (datalogger) should also be mentioned as well as

frequency of site visits for downloads.

P4,L4-6 – It would seem that you know that the probes are not well anchored in permafrost and the change in the "stickup" is due to heave rather than settlement of the ground in response to permafrost thaw (which might be the case if your probes extended to greater depths in the permafrost). Heave of the probe would take place over the winter as the freezing occurs and I assume you make the correction in the summer (although details are not provided). One might question how reliable your winter temperatures are in terms of the depth of measurement. More detail should probably be provided with respect to the amount of heave that occurs annually as well as how the temperatures are corrected.

P5, L11-12 – Do you correct for the vegetation effect? Trim the vegetation?

P5,L21 – revision suggested "...compile the dataset." P5,L23 – Does this mean that you might lose the 1m depth at sites where there has been significant heave of the probe? Minor revision suggested "...beyond the maximum observed depth..."

P5,L25-26 – revision suggested "...models are monthly, the monthly means were calculated for all variables, including air and....." P5, L26-27 – "Annual means were also calculated to allow...." Do you mean relationship between air and ground temperatures? P5, L31 – Is the frost number determined for only the ground surface temperature or at each depth? Also, do you include the freezing and thawing degree day indices in the dataset as these are useful for models etc.

P6, Eq 1-3 – For DDF are you using a complete winter/freezing season (e.g. Oct – May). You should probably provide a bit more explanation.

P6, L6 – revision suggested "Data records from many sites have gaps..." Also, equipment malfunction is another problem that may result in data gaps. One thing that is not mentioned is the frequency of site visits.

P6, L7-15 – For the missing data allowance, was there any consideration given to varying this according to the particular variable and its short-term variability. The deeper ground temperature would exhibit less variable so perhaps there could be allowance for more missing data than air temperature for example.

P6,L17 – P7,L7 – This section could probably be simplified and shortened. Maybe you could say that a unique name is assigned to each site. You could briefly mention how you deal with replacement sites.

P7,L11 – Effects of snow on ground thermal state – is this validation or analysis? P7,L21-22 – revision suggested "...keep the ground warm by reducing cooling (or heat loss) during the winter" P7,L23 – revision " ...snow depth and soil thermal properties." P7,L24 – There is no snow cover outside of Oct-Mar for even more northerly locations?

P7,L24-30 – This section is a bit confusing and more information/explanation should probably be provided especially since the parameters mentioned are specific to Slater et al. (2017) and may not be familiar to many readers (i.e. use parameters like n-factors, offsets to describe effect of snow etc.). How is SNDeff determined. Is it represented by one of the terms in Eq 4? Is SHTM equivalent to deltaAmpnorm? Is Amp-grnd referring to ground surface temperature, since snowcover will influence surface temperature, whereas the damping effect at depth will be more dependent on ground thermal properties.

P8, L11 – "spatially variable" better than "spatially complex" P8,L12 – delete last part of sentence "according to the synthesis dataset" – not necessary as it is shown in the figure that is derived from your dataset. P8,L13 – You could just say "located near the glacier" P8,L16 – revise "The other two sites,....." P8,L17-18 – Did you mean to include this last part of the sentence? You could make a comment that the thin snowcover is due to wind exposure. P8,L19 – This is the Frost Number calculated by Eq (1)? I don't see this value in table 3 only the freezing and thawing degree day indices.

P9,L1-14 – Wouldn't the comparison of trends for ground temperature at various depths be the most important thing to check for sensor drift etc. (i.e. ignore any snow effects

and focus on propagation of temperature wave with depth).

P10,L13-15 – See earlier comment regarding more explanation required for these parameters (SHTM, effective snow depth).

P11 Figure 6 – Labels on Y axis overlap between graphs. The trend requires correct units, degC/year, m/year. Are you showing standard error of the estimate also on the graph (should mention in caption)

P12, Figure 7 – Error bars represent standard error from the regression analysis?

P13,L1-4 – While snow is an important factor and influences the winter ground temperature (and therefore the amplitude), vegetation and ground cover can also effect the amplitude through their influence on summer temperature. Is this part of the reason for the considerable scatter in your graph?

P13,L5-6 – Delete this – repetitive.

P13,L8 – It is more correct to say "Changes in near-surface ground temperatures over time are important indicators of a changing climate" The direction of the change in ground temperature will indicate whether there is warming or cooling.

P13, L8-18 – Will the database be periodically updated as new data are collected? You mention it is worth maintaining but you could say more regarding plans for updates.

Table 1 – In section 2.2, interpolation to determine ground temperature for 4 target depths is mentioned. In the table, reference is made only to 1m. Are they any statistics calculated for the other depths? It isn't clear from the table or section 2.2.

---

## Referee Comment (RC2) · Anonymous Referee #2 · 11 Jun 2018

The dataset described in this manuscript is certainly useful, the manuscript, however, in its current form is not. There is too much ambiguity and missing information about the dataset, the language and organization are confusing in many places, and the presentation is a bit sloppy (especially the figures). This makes the utility of the dataset difficult to assess. The organization of the results section is strange and the paper leaves it unclear what all variables are actually included in this dataset and which ones are just presented for some type of qualitative validation.

Major points:

1. It is not clear what the dataset actually is. The introduction (p. 2 lines 30-31) says it is *measured* air and ground temperatures, snow depth, and soil volumetric water content. But then later other variables like frost numbers, thawing index, and freezing

index are mentioned. It is also later written that the data are provided as interpolated values (p. 5 lines 22-23), which is very different from measured values. At the very least, readers should come away from this paper knowing clearly what the dataset actually is.

2. A shortcoming of the dataset is the monthly timescale. While this may be OK for model validation, this is a very limited audience and most non-modelers would probably prefer the daily data. From a practical standpoint, you are disincentivizing users to turn to your synthesis product, given that the daily data are already readily available from the original UAF, USGS, and NPS sources. For example, based on the data you are providing I could not use it to quantify many processes that occur on shorter timescales, such as the onset of thaw or freeze, snowmelt timing, etc.

3. The dataset is supposed to be a synthesis of near-subsurface ground temperature data (p. 2 line 27). However, the "Overview of this dataset" section focuses on volumetric water content, snow depth, and frost number. It is very strange that you are providing an overview of some of the peripheral and derived variables, but not of the primary variables that make up the dataset. In fact, the entire final paragraph of section 3.1 should be deleted, because it suddenly presents research results, as opposed to describing and showcasing the dataset.

4. A crucial missing piece of this dataset is metadata information about the soil itself (soil type, density). This information is already available because, as stated (p. 5 line 4-5), at least the GI-UAF sensor installation was dependent on the soil profile and texture. Given the heterogeneity of soils and therefore its thermal conditions, adding this to the dataset would make it vastly more useful.

5. I question the linear interpolation method employed. Other studies, for example, Sherstiukov (2009) and Streletskiy et al. (2008, 2015) found that a polynomial fit better captures the exponential attenuation of temperature with depth. Regardless, this interpolation must be described with much greater detail if the entire dataset is based

on this interpolation. How many soil depth observations were required for the interpolation? Did you only interpolate between 'adjacent' soil depths? Or, for example, if a USGS location only had a 5 cm and a 1.2 m observation, did you still interpolate and provide temperature at 0.25, 0.5, 0.75, and 1 m? Was the interpolation done on the raw hourly or daily data, or on the final monthly data? Does the final, published dataset only contain these interpolated values, or also the original ones? Do you distinguish between observed and interpolated data in the official dataset? These are all crucial details that cannot be omitted.

6. Because this dataset is comprised of interpolated data (p. 5 line 22-23) anyway, why not also interpolate to fill in missing observations (or did you)? In cases where there are only one or two days of missing soil temperatures at a certain depth, you could relatively reliably fill in that gap based on the average of the previous and next day's observations. And if a certain soil depth has a missing observation but the immediately adjacent upper and lower depth observation is available, the missing depth could also be interpolated.

7. It is peculiar that a linear regression trend analysis was chosen as a core validation technique, given that you acknowledge how there are probably not enough data available to reliably do so. Using trend analysis and other derived variables like frost numbers and effective snow depth to then qualitatively 'eyeball' whether things generally look right are not robust validation techniques. This may uncover glaring issues, but not the subtle non-climatic artifacts and discontinuities that commonly plague climate data.

Specific points:

Can you quantify or estimate the thermal disturbance caused by drilling the holes at the soil temperature measurement sites?

p. 1 line 9: why report that the dataset consists of 41,667 monthly values (to make it seem really big)? This is not a useful statistic but instead you should provide a

percentage of how complete or how much missing data there are based on the overall date range.

p. 3 line 24-25: is this a personal hunch, or can you provide a reference for this statement?

p. 3 lines 28-29: please clarify how or why the thermistors are designed for a low temperature of -30°C, yet they record down to -35°C?

p. 5 lines 16-20: you explicitly mention the temporal availability of the USGS and GI-UAF data, but why not for NPS?

p. 6 lines 12-13 and 14-15: how did you choose those thresholds (20 days and 90%)? Based on those cited references, or did you perform your own cost-benefit analysis to determine how many missing observations you can allow while still obtaining the most continuous monthly/annual dataset?

p. 7 lines 5-6: what is this statement based on? Is this a visual assessment or did you perform a statistical analysis? Was this the only site that experienced a station move or other non-climatic change? Is there a list of all station moves, instrument changes, and other events that could affect the data quality?

p. 7 line 25: what is effective snow depth and how was it calculated?

p. 8 lines 21-24: you are reporting frost numbers without even explaining what those 0.5 versus 0.6 values mean. How is frost number used to indicate permafrost occurrence? What does a 0.5-magnitude frost number indicate?

p. 9 lines 5-7: how or why were those stations chosen? Are they the only ones with 10 or more years of data?

p. 10: is Smith Lake the only instance of multiple sites for the same (general) location? How or why was this Smith Lake example chosen for discussion?

Figure 5: there is enough room on these figures to write out the actual variable names

in the title.

Figure 6: primary y-axis labels overlap (why not write "Air (°C)," "Surface (°C)," "Ground (°C)" and elaborate in the caption that the top three rows show temperature?); I am unsure about the secondary y-axis labels, are they necessary? What does A, B, C, D refer to? What does the asterisk mean? What does the grey shading mean?

Figure 7: what is being shown here on these modified box and whisker plots? What does the circle versus the range indicate? Are all the trends plotted for the identical time period (the caption implies not)? If not, they are not comparable and this plot is misleading. Even if a location has more than 5 years of data, you cannot compare a 6-year trend to a 10-year trend if they have different beginning and end points. Are all the trends significant? What does A, B, C in the legend mean?

Figure 8: what is the grey shading?

Figure 9: needs actual x and y axis titles (instead of acronyms), and units.

The entire manuscript needs to be carefully edited. I am sure one of the 15 authors is a native English speaker who could do this?

---

## Referee Comment (RC3) · Anonymous Referee #3 · 3 Jul 2018

Overview:

This analysis provides an assessment of soil temperature, soil moisture, air temperature and snow depth data collected across Alaska, at depths up to 1 meter and over a time span of 1997 to 2016. The manuscript describes the processes used to harmonize the data; the harmonized data are presented in a resulting dataset posted through the Arctic Data Center. The dataset provides a useful contribution to the Arctic community and is especially relevant for model development. The manuscript itself could use more refinement prior to publication.

General comments:

Although the discussion appears focused on trends, the manuscript would benefit from

some analysis and discussion of interannual variability observed in the data.

Is it possible to calculate the start and end of the annual frozen period (where soil temperature is less than 0 degree C) from the compiled dataset? If so, please include this in the results and discussion, in addition to the freezing and thawing index.

A brief description of vegetation and soil type should be included for all sites, as well as mean annual thaw depth.

Specific comments:

Abstract.

Line 2. Some of these temperatures are at or above 0 degrees C and near-surface soil temperature and soil moisture are also included in the compiled dataset. Perhaps refer to these data as representing active layer and permafrost.

Line 6. Add a comma to 1327 meters for consistency.

Line 8. I don't think it necessary to have the paragraph mentioning missing data here, in the abstract. This is more a point to be made in the discussion and conclusion section.

Introduction.

Line 26. Remove "of" prior to "allow".

Line 35. Change "These technical validation would be useful for proving data harmonization and reusing these data" to "These technical validations are useful for data harmonization and future re-analysis of these data"

Section 2, Page 3.

Line 10. "Hydra" probe instead of "Hydro" probe?

Section 2, Page 5.

Line 11. Use "it is" instead of "it's"

Section 2.2, Page 5.

Line 14. You define QC here but do not consistently abbreviate quality control past this point. Do so for consistency, or just use "quality-control" instead of QC.

Line 22. Was bias introduced when applying linear interpolation to ground temperatures? The use of linear interpolation needs to be justified.

Line 32. Add another sentence or two to describe the Frost index in more detail. Provide an example of how the resulting index values might be interpreted.

Section 3.1, Page 8.

Line 12. Southeast boreal or southeast mountain tundra?

Line 13. In which year was the 1.5 m depth recorded?

Line 17. I do not think the note in parentheses is necessary.

Section 3.2, Page 10.

Line 1. Change "while" to "that". Add "was" before "mainly".

Lines 10 & 11. Change Smith Lake to SL2 and SL3.

Section 3.2, Page 13.

Lines 5-6. These sentences are not necessary.

Figure 1.

Move the legend for the pan-Arctic permafrost inset to the right-hand side of the inset map. Increase the font size – otherwise some will not be able to read this. Increase the font size for the main legend.

Figure 2. Why show only snow depth? Why not also show soil moisture, air and ground

temperature? Indicate spatial locations where the "trend" analysis shows significant change or no significant change (I realize that locations having >= 10 years of data may be limited, but it is still helpful to see these on a map). Color code by p value?

---

## Author Comment (AC1) · 10 Oct 2018

The authors thank all three anonymous referees and the editor Dr. Chris DeBeer for their constructive comments and suggestions, which are very important for us to improve the present manuscript and dataset. We have addressed each in turn, and hope that our responses are adequate. Figures and tables were also updated accordingly. Please see the attached *.pdfs for our responses to the reviewers and revised manuscript.

Please also note the supplement to this comment:
https://www.earth-syst-sci-data-discuss.net/essd-2018-54/essd-2018-54-AC1-
supplement.zip